# Global Trends and Correlates of COVID-19 Vaccination Hesitancy: Findings from the iCARE Study

**DOI:** 10.3390/vaccines9060661

**Published:** 2021-06-17

**Authors:** Jovana Stojanovic, Vincent G. Boucher, Myriam Gagne, Samir Gupta, Keven Joyal-Desmarais, Stefania Paduano, Ala’ S. Aburub, Sherri N. Sheinfeld Gorin, Angelos P. Kassianos, Paula A. B. Ribeiro, Simon L. Bacon, Kim L. Lavoie

**Affiliations:** 1Department of Health, Kinesiology & Applied Physiology, Concordia University, Montréal, QC H4B 1R6, Canada; jovana.stojanovic@mail.concordia.ca (J.S.); joyal008@umn.edu (K.J.-D.); simon.bacon@concordia.ca (S.L.B.); 2Montreal Behavioural Medicine Centre, Research Centre, Centre Intégré Universitaire de Santé et de Services Sociaux du Nord de l’Ile de Montréal (CIUSSS-NIM), Montréal, QC H4J 1C5, Canada; vincent.gosselin.boucher@gmail.com (V.G.B.); paulaabribeiro@gmail.com (P.A.B.R.); 3Department of Psychology, University of Québec at Montreal (UQAM), Montréal, QC C3H 3P8, Canada; 4Unity Health Toronto, Department of Medicine, Division of Respirology, St. Michael’s Hospital, Toronto, ON M5B 1W8, Canada; myriam.gagne@unityhealth.to (M.G.); samir.gupta@unityhealth.to (S.G.); 5Keenan Research Center, Li Ka Shing Knowledge Institute, St. Michael’s Hospital, University of Toronto, Toronto, ON M52 3M2, Canada; 6Section of Public Health, Department of Biomedical, Metabolic and Neural Sciences, University of Modena and Reggio Emilia, 41125 Modena, Italy; stefania.paduano@unimore.it; 7Physical Therapy Department, Faculty of Allied Medical Sciences, Isra University, Queen Alia International Airport South of the Capital Amman, Amman 11622, Jordan; ala.aburub@mail.mcgill.ca; 8Department of Family Medicine, University of Michigan School of Medicine, Ann Arbor, MI 48109, USA; sherri.gorin@gmail.com; 9Department of Psychology, University of Cyprus, Nicosia 2109, Cyprus; kassianos.angelos@ucy.ac.cy; 10Department of Applied Health Research, University College London, London WC1E 7HB, UK

**Keywords:** COVID-19, vaccine hesitancy, international analysis, cross-sectional survey

## Abstract

The success of large-scale COVID-19 vaccination campaigns is contingent upon people being willing to receive the vaccine. Our study explored COVID-19 vaccine hesitancy and its correlates in eight different countries around the globe. We analyzed convenience sample data collected between March 2020 and January 2021 as part of the iCARE cross-sectional study. Univariate and multivariate statistical analyses were conducted to explore the correlates of vaccine hesitancy. We included 32,028 participants from eight countries, and observed that 27% of the participants exhibited vaccine hesitancy, with increases over time. France reported the highest level of hesitancy (47.3%) and Brazil reported the lowest (9.6%). Women, younger individuals (≤29 years), people living in rural areas, and those with a lower perceived income were more likely to be hesitant. People who previously received an influenza vaccine were 70% less likely to report COVID-19 vaccine hesitancy. We observed that people reporting greater COVID-19 health concerns were less likely to be hesitant, whereas people with higher personal financial concerns were more likely to be hesitant. Our findings indicate that there is substantial vaccine hesitancy in several countries, with cross-national differences in the magnitude and direction of the trend. Vaccination communication initiatives should target hesitant individuals (women, younger adults, people with lower incomes and those living in rural areas), and should highlight the immediate health, social and economic benefits of vaccination across these settings. Country-level analyses are warranted to understand the complex psychological, socio-environmental, and cultural factors associated with vaccine hesitancy.

## 1. Introduction

In 2019, low vaccine intentions or vaccine hesitancy, defined as a ‘delay in acceptance or refusal of vaccination despite availability of vaccination services’ was reported as one of the top ten threats to global health by the World Health Organization [1,2]. Hesitancy is a complex and dynamic problem, driven by individual and group factors, context-specific issues, and vaccine-specific influences [3,4]. The impact of these factors can be hard to predict within unstable environments such as pandemics and epidemics.

Since the beginning of the SARS-CoV-2 pandemic, major public health prevention efforts, such as social isolation, contact tracing, masks, and lockdowns, have been implemented throughout affected countries to lessen the unprecedented health and social impacts [5]. Importantly, significant investments were made by several countries to rapidly develop and test SARS-CoV-2 vaccines. By the beginning of 2021, seven vaccines were approved in various countries, and around half a million doses were administered globally. Since then, the rollout has substantially increased—as of April 2021, over 12 million people had received at least one dose worldwide [6,7,8].

Beyond the complex logistical issues around mass manufacturing, global access and distribution [9], COVID-19 vaccine hesitancy among the public might threaten governments’ hopes of achieving herd immunity [10]. Despite initial data showing promising vaccine uptake among those who are willing [6], substantial work needs to be done moving forward in order to incentivize those who are ambivalent and hesitant to receive the vaccine. This has proved to be difficult in situations with an evolving epidemiological situation [11], the emergence of novel viral variants [12,13], and a continuous inflow of new vaccine-related information (including complications) [14]. Public health institutions are struggling to combat emergent factors influencing the surge in vaccine hesitancy, many of which are unique to the current pandemic-related circumstances, such as concerns about the expedited process of vaccine development, testing, and approval, the vaccine and COVID-19 infodemic (including a great deal of false information), and the politicization of vaccines [15].

To date, an increasing number of studies have shown that COVID-19 vaccine hesitancy can range from less than 10% to an overwhelming 40–50%, with great variability across different countries and sub-populations [16,17,18]. However, there is limited availability of single-study data that enable direct cross-country comparisons of COVID-19 vaccine hesitancy, using a consistent formulation of the hesitancy question over time [16,18].

Understanding and continuously assessing COVID-19-specific predictors of vaccine hesitancy could help prioritize communication targets, and help tailor the message format and content, with the goal of better addressing the underlying reasons for vaccine hesitancy and instilling trust in the public. With this aim in mind, we provide a map of this issue, assess changes over time, and evaluate the diverse correlates of vaccine hesitancy across eight countries. This is the first study to describe COVID-19 vaccine hesitancy using the consistent measure of hesitancy throughout seven waves of survey data collection, across three different continents.

## 2. Materials and Methods

### 2.1. Study Design

The iCARE Study (www.icarestudy.com (accessed on 14 June 2021)) is an ongoing international cross-sectional survey series that uses different survey recruitment approaches to capture public awareness and attitudes towards COVID-19 and related public health measures. The iCARE study is led by researchers at the Montreal Behavioural Medicine Centre (MBMC: https://mbmc-cmcm.ca/covid19/ (accessed on 14 June 2021)). The details and methodological background of the iCARE study have been published elsewhere [19].

### 2.2. Study Participants and Recruitment

We report data from the iCARE global convenience sample from 27 March 2020 to 31 January 2021. The iCARE survey (distributed using LimeSurvey© from Montreal, Canada) was administered in 40 countries around the world using an online snowball sampling strategy, through the direct engagement of the study collaborators. The survey distribution occurred through professional associations and societies, university networks, community organizations and groups, social media, and personal contacts.

To ensure sufficient global representation with consistency over time, we divided the study timeline into three different periods, based on the global COVID-19 epidemiological situation: period 1: 27 March–31 May 2020; period 2: 1 June–15 September 2020; and period 3: 16 September 2020–31 January 2021 (Appendix A). Eight countries (Brazil, Canada, Colombia, France, Italy, Turkey, the United Kingdom (UK), the United States of America (USA)), with a total of 32,028 participants were included in the analysis.

The study was approved by the Research Ethics Committee at the Centre intégré universitaire de santé et de services sociaux du Nord-de-l’Île-de-Montréal (CIUSSS-NIM) (REB#: 2020-2099/03-25-2020). The present paper is presented in line with the Strengthening the Reporting of Observational Studies in Epidemiology (STROBE) statement (Appendix A) [20].

### 2.3. iCARE Survey Questionnaire

The survey was designed to measure constructs related to the COM-B Model [21], which predicts that behaviour change depends on the awareness of prevention measures (capability), individuals’ beliefs that measures are personally relevant and important (motivation) and social and environmental structures that enable the required behaviours (opportunity); and the Health Beliefs Model [22], which predicts that behaviour change depends on individuals’ belief in the personal threat(s) of the disease, as well as their beliefs around how important and effective the recommended behaviour is. Survey questions included the following: sociodemographic variables; physical/mental health; prior COVID-19 infection; general health behaviours; awareness of local government and municipal policies; perceptions and attitudes about these policies; concerns about the virus and its impacts; behavioural responses; and vaccine intentions (available online: https://osf.io/nswcm/ (accessed on 14 June 2021)). A detailed map of the survey questions to the theoretical behaviour change frameworks is provided elsewhere [19]. COVID-19 vaccine intentions were assessed using the question “If a vaccine for COVID-19 were available today, what is the likelihood that you would get vaccinated?” (possible responses: Extremely likely, Somewhat likely, Unlikely, Very unlikely, and I don’t know/prefer not to answer). All the questionnaire items that included an ‘I don’t know/I prefer not to answer’ response were considered missing values. In our analyses, we dichotomized this variable into “Extremely likely” (not vaccine hesitant) vs. all others (vaccine hesitant).

COVID-related concerns were measured with different survey items, with the following possible answers: To a Great Extent, Somewhat, Very Little, and Not at All.

### 2.4. Data Analysis

Descriptive statistics were calculated to provide an overview of the study sample in terms of their sociodemographic characteristics and selected lifestyle habits, according to the vaccine intentions response. Additional graphical presentations are provided to present changes in vaccine intentions between different countries and across time points. All questionnaire items that included an ‘I don’t know/I prefer not to answer’ response were considered missing values, and statistical analyses were based on complete case records.

We applied univariate and multivariate logistic regressions in order to evaluate the association between vaccine hesitancy (dependent variable) and a series of demographic and health factors (independent variables), which were pre-selected based on the previous literature (age, sex, perceived income, education, continent, residential area (i.e., urban, suburban and rural), the presence of chronic health conditions, and influenza vaccination history) [4]. The Cochran–Armitage test for trends was used to assess whether a trend in vaccine hesitancy was present across ordered periods of time, both in the overall sample and across individual countries.

To cluster the COVID-19-related concerns, we performed a principal component analysis (PCA) with varimax rotation on a polychoric correlation matrix of 11 items that were assessed in every survey of the iCARE study. We identified four components based on the Kaiser criterion (eigenvalue > 1.0), scree plot, and components’ interpretability [23]. Items with component loadings higher than 0.4 were used to interpret each component of the COVID-19 concerns. The four components were: ‘Health concerns (self)’, including concerns about being infected and the impact of infection on one’s health, ‘Health concerns (others)’, including concerns about infecting close individuals and people in the community; ‘Personal financial concerns’, including concerns about losing income and inability to pay food or housing; ‘Social/economic concerns’, including concerns around the country entering an economic recession and how long it will take to return to normal (see Appendix A for a list of items and component loadings). A separate multivariate logistic model was applied to test the associations between vaccine hesitancy (dependent variable) and these four COVID-19-related concern variables, with adjustments for age, sex, education, residential area, continent, and survey period. All statistical tests were two-sided, and a *p*-value < 0.05 was considered statistically significant. The statistical analysis was performed using SAS, version 9.4.

## 3. Results

### 3.1. Sample Description

Our sample included 32,028 responses from eight countries across three different continents (North America, South America, and Europe). The majority of individuals were female (72.2%), between 30 and 64 years of age (60.6%), with a graduate or postgraduate degree (77.3%) and in the middle third of perceived income (54.1%) (Table 1, Appendix A). Around two-thirds of respondents lived in urban areas. The sociodemographic characteristics of our samples, broken down by time and individual country, can be found in Appendix A.

### 3.2. Estimates of Vaccine Hesitancy and Changes over Time

Overall, almost 27% of our sample reported vaccine hesitancy (Appendix A). We observed significant increases in hesitancy levels across our entire sample over time (period one: 25.6%; period two: 27.5%; period three: 29.9%, *p* < 0.0001 for trend, see Figure 1). When looking at individual countries, French residents reported the highest rate of vaccine hesitancy (39.9%, 51.2%, and 50.9% across period one, two, and three, respectively) and Brazilian residents reported the lowest (8.4%, 9.5%, and 11.1% in period one, two, and three, respectively). Italy reported low proportions when the country was hardest hit during the initial period of the pandemic (9.3%), but hesitancy then increased to around 19% in period 3 (*p* < 0.0001 for trend). Significant trends in increased vaccine hesitancy were also observed in Canada, Colombia, France, Turkey and the USA (data not shown). Conversely, there was a non-significant trend towards a decrease in hesitancy from 26.7% in period 1 and 27.6% in period 2, to 20.3% in period 3 among UK residents (*p* = 0.35 for trend).

### 3.3. Predictors of Vaccine Hesitancy

Table 2 presents the findings from the univariate and multivariate analyses examining the association between vaccine hesitancy and the sociodemographic and health characteristics of individuals. In the following section, we provide a summary of the multivariate analysis. Males, older adults and urban-dwelling individuals were less likely to report COVID-19 vaccine hesitancy (OR = 0.84, 95% CI = 0.78–0.91 for males compared to females; OR = 0.75, 95% CI = 0.64–0.88 for individuals over 65 compared to under 29 years old; and OR = 0.83, 95% CI = 0.74–0.92 for individuals living in urban compared to rural areas). Furthermore, compared to people with the lowest perceived income, those reported as being in the middle or top tertile for income expressed a lower likelihood of vaccine hesitancy (OR = 0.81, 95% CI = 0.73–0.90 and OR = 0.53, 95% CI = 0.47–0.59 for middle and top income tertiles compared to the lowest tertile, respectively). Those reporting a history of influenza vaccination were 70% less likely to report any level of COVID-19 vaccine hesitancy (OR = 0.26, 95% CI = 0.24–0.29). Having a chronic health condition was significantly associated with lower hesitancy in the univariate analysis, but these effects were attenuated in the multivariate models. In our multivariate modelling analysis, we clustered the countries into continents and observed that, compared to individuals from North America, individuals from Europe were 37% more likely and those from South America were 35% less likely to report vaccine hesitancy.

### 3.4. Association between COVID-19-Related Concerns and Vaccine Hesitancy

In line with the constructs of the Health Beliefs and COM-B models, we evaluated the association between COVID-19-related concerns and vaccine hesitancy in a multivariate analysis adjusted for age, sex, education, living area, continent, and survey period. Our analysis showed that people with higher personal health concerns and higher health concerns for other individuals were significantly more likely not to report hesitancy (β = −0.374, *p* < 0.001; and β = −0.309, *p* < 0.001, respectively) (Table 3). In contrast, people with higher levels of personal financial concerns were more likely to be hesitant (β = 0.369, *p* < 0.001). Lastly, having higher social/economic concerns was significantly associated with not being hesitant (β = −0.199, *p* < 0.001).

## 4. Discussion

We used a multi-round cross sectional survey to understand the trends in COVID-19 vaccine hesitancy in over 32,000 individuals from eight countries. We further sought to identify factors associated with vaccine hesitancy as a function of sociodemographic characteristics and reported COVID-19-related concerns.

Our data show that COVID-19 vaccine hesitancy has increased over time across seven countries in our sample, which may signal a troubling trend globally. Although most countries are in the early to mid-phases of their vaccine campaign, and demand exceeds supply in most settings, given the estimated required population vaccination threshold of 70–80% for herd immunity, our findings suggest that a focus on vaccine hesitant individuals will be required for the global vaccination effort to ultimately succeed. This can be particularly troubling for countries such as France and Turkey, which reported the highest hesitancy estimates across all three time points, in line with the previously published literature [4,24]. On the contrary, the Brazilian population was the most vaccine receptive in our dataset, with hesitancy levels ranging from only 8 to 11%. These estimates are somewhat comparable to representative sample data reported by Lazarus et al. in June 2020, showing high acceptance across Latin American countries (including 85% vaccine acceptance in Brazil) [25]. Moreover, the UK was the only country in our sample that reported promising decreases in hesitancy over time. This positive turnaround was also observed in a longitudinal analysis from England and Wales, showing that almost four in five adults who were hesitant in December 2020 planned to receive or had already obtained the vaccine in February 2021. This shift can be attributed to the early vaccine availability, aggressive vaccination campaign and public health communication efforts [26]. Our analyses demonstrated a high degree of variability in vaccine hesitance across countries when measured at the same time and with the same measures. This likely reflects the complex interplay between social, cultural, demographic, and public health-related determinants of vaccine hesitancy, and requires further study.

We identified several sociodemographic correlates of poor COVID-19 vaccine intentions. Certain demographic characteristics were associated with higher hesitancy (being female, of a younger age, living in a rural area, and having a lower perceived income). This is largely in line with the published COVID-19 literature and with evidence from previous pandemics [16,18,25,27,28,29]. We observed that women, compared to men, experienced greater hesitancy levels, similar to the growing body of COVID-19 literature [30,31]. Our finding is somewhat surprising, especially considering that women continuously reported better adherence to COVID-19 preventive behaviours, most likely due to increased risk perceptions [32,33] and greater concerns about the negative health and social impacts of the virus [34,35]. However, sex differences in vaccination outcomes have been documented previously, with women reporting higher antibody responses [36] and more adverse events following vaccination for influenza, hepatitis B, and yellow fever vaccines [36,37]. Relative to the COVID-19 vaccines, rare and serious side effects were more prevalent among females in two widely used vaccines, which may have led to increased levels of worries and safety concerns among this population [35,36,38,39,40].

Our data showed that respondents over 65 years of age had 25% lower odds of hesitancy, compared to individuals under 30. Higher levels of risk perception and actual risk of getting sick or developing serious complications from COVID in older individuals most likely explain these differences [41]. Almost 30% of younger and middle-aged adults reported some levels of hesitancy, which might significantly undermine vaccine rollout efforts in community-dwelling adults. Moreover, our analyses demonstrated that there are socioeconomic disparities in vaccine acceptance. Individuals with a lower perceived income reported poor vaccine intentions; however, they are at increased risk of acquiring the infection, as well as suffering from negative direct or indirect consequences of COVID-19 [5,42]. This suggests that there is an urgent need for governments and health authorities to study the determinants of hesitancy in these groups more closely, and to develop correspondingly targeted interventions for these populations. This will require investments in areas such as increasing vaccine and overall health literacy, as well as acknowledging previously identified sources of vaccine hesitancy in these populations, such as distrust of healthcare systems and authority [3]. In many settings, these individuals are also more likely to be from racialized and new immigrant communities, suggesting the need for culturally and linguistically sensitive vaccination messages. Lastly, our data demonstrated a strong influence from past behaviours, with individuals reporting a previous influenza inoculation having 70% lower odds of hesitancy, compared to individuals that did not. These findings might reflect the fact that individuals’ previous positive experiences with influenza vaccines may have resulted in them having more reassuring attitudes towards COVID vaccination. To support this, findings from previous [28,29] and current pandemics [43,44,45,46,47,48] showed that regular influenza vaccine takers reported increased risk perception regarding H1N1 influenza, as well as positive attitudes in regard to vaccine safety and the value of vaccination in general.

Population-specific concerns and motivations may be leveraged to create targets for vaccine hesitancy communication strategies. A large body of literature suggests that vaccination decisions may depend on perceptions about individual and community risk [49]. This is consistent with both the Health Beliefs and COM-B models, which assume that greater concerns about the health consequences of the virus (on an individual, friends, family, and the community) can influence both the intention and adoption of disease prevention measures, such as vaccination. In reference to this, we evaluated the relationship between different types of COVID-19-related concerns and vaccine hesitancy. Our results demonstrated that individuals with higher levels of personal health concerns were less likely to report vaccine hesitancy, consistent with prior studies carried throughout the countries [27,43,44,50,51]. Moreover, elevated concerns about the health of other individuals, including family and friends, were significantly associated with lower vaccine hesitancy in our study. These findings demonstrate that communications should stress the social benefits of vaccination, including the protection of close individuals and society as a whole [52]. Finally, our multivariate modelling revealed that people with higher personal financial concerns were more likely to demonstrate vaccine hesitancy, which parallels previous iCARE data showing that adherence to other COVID-19 preventive behaviours correlates with individuals’ financial situation and concerns [35]. Importantly, considering the negative impacts of the current pandemic on the broader economies of societies and individuals, our results suggest that communication strategies for increasing vaccine uptake should address the economic benefits of vaccination, including positive economic impacts on individual families (i.e., a reduced need for further societal lockdowns, affecting businesses, and the personal ability to avoid reduced productivity at work and a loss of pay due to illness and recovery) [53]. Reflecting upon the COM-B model, which stresses the importance of having suitable social and environmental resources for behaviour change, it is likely that the financial concerns evaluated in our study represent an important aspect of the ‘reflective motivation’ for enabling the adoption of the vaccine behaviour.

Our study has several limitations. We employed a multi-round cross-sectional survey, which enabled us to provide snapshots across different periods of time, but direct causality between the variables of interest and vaccine hesitancy cannot be inferred. Data were collected using a convenience sample methodology, with an overrepresentation of women and highly educated individuals. The outputs may not be representative of the general population in the individual countries. However, our data might be underestimating the extent of hesitancy in the general population in these countries. Unpublished data from the iCARE representative sample reported higher figures of hesitancy in the Canadian population, ranging from 35% in April to 49% in November, 2020 [54]. Furthermore, our data were collected up to January 2021, a time point in which many of the countries began their vaccination campaigns, which might have affected overall hesitancy rates in the population. Nevertheless, the predictors of hesitancy are unlikely to change dramatically and our analysis still provides a way forward with regard to which groups to target and how. Moreover, we acknowledge the absence of validated vaccine hesitancy measures and the fact that the iCARE questionnaire measured participants’ intention to receive a vaccination, which overestimates actual vaccination behaviour, though it is noted that intentions are among the best predictors of actual behaviour [55]. Lastly, we did not analyze the influence of certain other predictors, such as trust in different information sources and the government, as these questions were not assessed in each version of the survey.

## 5. Conclusions

In summary, global vaccine hesitancy increased between the beginning of the pandemic, in March 2020, and the period immediately before the wide release of vaccines, in January 2021. Hesitancy varied across the eight countries in our sample, and temporal trends showed increases in vaccine hesitancy for the majority of the studied countries. Future analyses should explore how the complex social, environmental, psychological, and cultural differences between countries may influence vaccine hesitancy. We also noted that those of the female sex, those of a younger age, those who live in rural areas, and those with a lower perceived income were more likely to be vaccine hesitant, as were those with lower perceptions of their own and others’ COVID-19-related health risks and those who reported higher degrees of personal financial concern. These participants should be key target audiences for global vaccination campaigns, and the findings indicate the need to address population-specific concerns around health and finances in vaccine communications.

## Figures and Tables

**Figure 1 vaccines-09-00661-f001:**
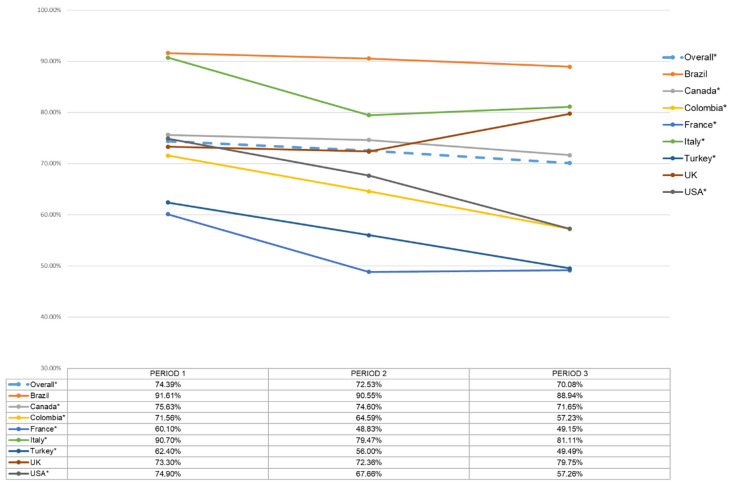
Positive vaccine intentions over time and across countries (defined as individuals that are Extremely likely to receive the vaccine). * indicates a significant test for a trend using the two-sided Cochran–Armitage trend test.

**Table 1 vaccines-09-00661-t001:** Sociodemographic and health characteristics of the sample as a function of vaccine hesitancy.

Sociodemographic and Health Variables	Vaccine Hesitancy				
	Somewhat Likely, Unlikely, Very Unlikely		Extremely Likely		*p*-Value ^a^
	N	%	N	%	
Overall	7074	26.61	19,508	73.39	
Missing values	5446				
Period ^b^					
3	1064	29.92	2492	70.08	<0.0001
2	1678	27.47	4431	72.53	
1	4332	25.61	12,585	74.39	
Continent					
North America	4022	25.66	11,653	74.34	<0.0001
Europe	2249	32.19	4738	67.81	
South America	803	20.48	3117	79.52	
Sex					
Women	5272	27.66	13,788	72.34	<0.0001
Men	1724	23.6	5581	76.4	
Age					
Less than or equal to 29	1898	28.32	4803	71.68	<0.0001
30–64 years	4510	28.18	11,495	71.82	
65 years or more	633	16.74	3148	83.26	
Education level					
High school or lower	1477	25.83	4242	74.17	0.3451
Graduate or postgraduate degree	5177	26.45	14,395	73.55	
Current employment status					
Unemployed	1095	21.26	4056	78.74	<0.0001
Employed	4135	27.72	10,784	72.28	
Student	550	32.05	1166	67.95	
Residential area					
Rural or country area	987	29.79	2326	70.21	<0.0001
Suburban or regional	1922	27.88	4972	72.12	
Urban or city	3753	24.79	11,385	75.21	
Perceived average annual household income					
Bottom third	1027	33.56	2033	66.44	<0.0001
Middle third	3433	27.82	8907	72.18	
Top third	1497	19.97	6001	80.03	
Health condition at risk					
No	4650	27.94	11,991	72.06	<0.0001
Yes	2071	23.23	6844	76.77	
History of seasonal influenza vaccination					
Never or once or twice in the last 5 years	5365	34.89	10,010	65.11	<0.0001
Every year and 3 times in the last 5 years	1188	12.41	8387	87.59	

^a^*p*-value from χ^2^ test for bivariate comparisons; ^b^ Period 1 (March–May 2020); 2 (June–15 September 2020); Period 3 (15 September 2020–January 2021).

**Table 2 vaccines-09-00661-t002:** Univariate and multivariate associations between sociodemographic and health characteristics of participants and vaccine hesitancy.

Sociodemographic and Health Variables	Univariate Analysis ^a^				Multivariate Analysis ^b^			
	OR ^c^	95% Confidence Interval		*p*-Value	OR ^c^	95% Confidence Interval		*p*-Value
		Lower	Upper			Lower	Upper	
Period ^d^								
3	1				1			
2	0.89	0.81	0.97	**0.0099**	0.77	0.69	0.86	**<0.0001**
1	0.81	0.74	0.87	**<0.0001**	0.63	0.57	0.7	**<0.0001**
Continent								
North America	1				1			
Europe	1.38	1.29	1.46	**<0.0001**	1.37	1.27	1.49	**<0.0001**
South America	0.75	0.69	0.81	**<0.0001**	0.66	0.58	0.74	**<0.0001**
Sex								
Women	1				1			
Men	0.81	0.76	0.86	**<0.0001**	0.84	0.78	0.91	**<0.0001**
Age								
Less than or equal to 29	1				1			
30–64 years	0.99	0.93	1.06	0.8243	1.04	0.93	1.15	0.5273
65 years or more	0.51	0.46	0.56	<0.0001	0.75	0.64	0.88	0.0004
Education level								
High school or lower	1				1			
Graduate or postgraduate degree	1.03	0.97	1.11	0.3472	1.07	0.98	1.18	0.1306
Current employment status								
Unemployed	1				1			
Employed	1.42	1.32	1.53	**<0.0001**	1.03	0.93	1.15	0.5495
Student	1.75	1.55	1.97	**<0.0001**	1.08	0.92	1.27	0.3643
Residential area								
Rural or country area	1				1			
Suburban or regional	0.91	0.83	0.99	**0.045**	0.98	0.87	1.09	0.6787
Urban or city	0.78	0.72	0.84	**<0.0001**	0.83	0.74	0.92	**0.0003**
Perceived average annual household income								
Bottom third	1				1			
Middle third	0.76	0.70	0.83	**<0.0001**	0.81	0.73	0.90	**<0.0001**
Top third	0.49	0.45	0.54	**<0.0001**	0.53	0.47	0.59	**<0.0001**
Health condition at risk ^e^								
No	1				1			
Yes	0.78	0.74	0.83	**<0.0001**	0.97	0.90	1.05	0.4244
History of seasonal influenza vaccination								
Never or once or twice in the last 5 years	1				1			
Every year and 3 times in the last 5 years	0.26	0.25	0.28	**<0.0001**	0.26	0.24	0.29	**<0.0001**

^a^ A series of univariate logistic regressions was conducted to assess the relationship between vaccine hesitancy and individuals’ sociodemographic and health factors (probability modeled is ‘Somewhat likely, Unlikely, Very unlikely’); ^b^ multivariate logistic regression was conducted to assess the relationship between vaccine hesitancy and sociodemographic and health factors (probability modeled is ‘Somewhat likely, Unlikely, Very unlikely’); analysis was done on 19,546 individuals due to missing values in either response or explanatory variables; Goodness-of-Fit Test and Hosmer and Lemeshow Goodness of Fit Test (*p* = 0.7783); McFadden’s index = 0.09; ^c^ OR—odds ratio; ^d^ Period 1 (March–May, 2020); Period 2 (June–15 September 2020); Period 3 (15 September 2020–January 2021); ^e^ Health condition at risk includes: any heart disease or history of heart attack or stroke, any chronic lung disease; active/current cancer; hypertension; diabetes; severe obesity; any autoimmune disease.

**Table 3 vaccines-09-00661-t003:** Multivariate logistic regression model estimating the association between vaccine hesitancy and COVID-19 related concerns.

				OR ^d^	95% CI ^e^	
Variable	Estimate ^a^	SE ^b^	*p*-Value ^c^		Lower	Upper
Intercept	1.101	0.108	<0.0001			
Health concerns (others) (continuous)	−0.309	0.025	<0.0001	0.73	0.70	0.77
Health concerns (self) (continuous)	−0.374	0.022	<0.0001	0.69	0.66	0.72
Personal financial concerns (continuous)	0.369	0.020	<0.0001	1.45	1.39	1.50
Social/economic concerns (continuous)	−0.199	0.022	<0.0001	0.82	0.79	0.86
Goodness-of-Fit Test ^f^ (*p* = 0.06)						

^a^ The model was adjusted for sex, age, education, area of living, continent, and period of survey. Probability modeled: ‘Somewhat likely, Unlikely, Very unlikely’. Analysis was conducted on 24,699 individuals due to missing values in either response or explanatory variables; ^b^ SE—standard error; ^c^
*p*-values for the chi-square test, testing the null hypothesis that the individual predictor’s regression coefficient equals zero, given that the other predictor variables are in the model; ^d^ OR—odds ratio; ^e^ 95% confidence interval for the regression parameters; ^f^ Hosmer and Lemeshow Goodness of Fit Test.

## Data Availability

The iCARE study data are available on request via the process identified here: https://mbmc-cmcm.ca/covid19/apl/ (accessed on 14 June 2021).

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
