# Peer review of "Global Trends and Correlates of COVID-19 Vaccination Hesitancy: Findings from the iCARE Study"

_vaccines, 2021, doi:10.3390/vaccines9060661_

Round 1

Reviewer 1 Report

I would like to congratulate the authors on this highly relevant manuscript, which describes a timely and well-executed study. If the authors would consider clarifying the points below, I believe this could strengthen their work:

  1. page 2, lines 61-69: in their argument, the authors focus on shifts in public sentiments, and underlying causes. However, I believe that governments (and, to a lesser extent, health institutions) have also shown shifts in ‘sentiment’ around vaccination complications, e.g., towards the safety of the AstraZeneca vaccine. In some countries, this vaccine was banned for certain age groups, and then later this ban was relieved again, et cetera. I think these shifts help explain shifts in ‘public sentiment’.
    More generally, I would suggest to stay away from the notion that publics can be sentimental, emotional, inconsistent, or in any other way (more) irrational as compared to governments, scientists, the pharmaceutical industry, or any other stakeholder for that matter.

  2. page 2, line 74: the term ‘harmonized framing’ is somewhat ambiguous. I believe the authors mean something like ‘consistent formulation’ (of the hesitancy question). Framing can mean a lot of things, both inside and outside the social sciences.

  3. page 3, lines 109-110: It is stated that the “The survey was designed to measure constructs related to the COM-B Model [18] and 109 Health Belief Model [19] (...)”; however, in the results the constructs from these models are not mentioned, at least not with explicit referral to these models. I would be especially interested in these constructs, as they can provide quite a comprehensive understanding of health behavior, and substantially explain variance, especially beyond demographics. To what extent where these construct indeed measured related to vaccine hesitancy, and, if these were measured, can these relationships be reported in a theoretically more explicit manner? This, I believe, would add to the scientific relevance of the findings, as well as help build the evidence base for these models.

  4. page 3, lines 118-119: It is stated that “All the questionnaire items that included an answer ‘I don’t know/I prefer not to answer’ were considered missing values.” Although I understand this decision, less clear is how missing values are treated in the analyses. As table 1 shows, the number of missing values is quite substantial. Could the authors explain how missing values were treated in the analyses? I also believe this can be described more transparently in section 3.1 (sample description).

  5. page 3, lines 115-121: Two aspects of how vaccine hesitancy is measured warrant elaboration:
    1. The item asks respondents “What is the likelihood that you would get vaccinated?” However, likelihood may also include perceptions of control, such as self-efficacy (social cognitive theory), capability (COM-B model), or opportunity (COM-B model). Also, it is only a single item to measure quite a complex concept such as intention.
    2. Second, on page 1, lines 43-44, vaccine hesitancy is defined as “delay in acceptance or refusal of vaccination despite availability of vaccination services”. This is clearly a behavior and not an intention, at least not exclusively. Considering these two points, could the authors give a more thorough rationale of the validity of the vaccine hesitancy measure? Also, was a behavioral measure of vaccine hesitancy also available, or at least considered? Some respondents could have reported that they indeed already refused the vaccine.

  6. Related to the Results, and in particular to Table 1 (pp. 5-6): what are the explained variances of the multivariate logistic regression models? Explained variance in vaccine hesitancy is an important indicator of how well the final model is able to explain variance in this dependent variable, when all predictors are included in the full model. This also gives way to a discussion of the performance of the model.

  7. Related to the study design and its limitations, and because different cross-sectional samples were used to discover trends over time, I was wondering whether the demographics, and thus representativeness and comparability, of the samples significantly changed over time?

  8. Page 10, lines 152-153: In the ‘Conclusions’ it is stated that “temporal trends were inconsistent”; however, this seems inconsistent with the Abstract (page 1, lines 27-28) that reports “increases over time”. Indeed, only the UK seemed to deviate from the trend of increasing hesitancy that is observed in the other countries?

  9. At the beginning of the Results and Discussion sections, the ‘author guidelines’ are stated for each of these sections. These can probably be deleted.

Reviewer 2 Report

Dear Authors,

I have read your paper with great interest. The finding that in the UK the phenomenon of vaccine hesitancy is decreasing is very important. However, I have few remarks:

  1. Decide for one spelling of COVID-19 - look for example page 2 line 80
  2. iCARE study survey was collecting data from 40 countries, you decided to present only data from 8 countries. Why? (page 2, line 94; page 3 line 101. Please explain.
  3. Please provide Response Rate
  4. Table 1 is really difficult to read due to a lot of data in one place. I would suggest splitting this information, which would make this table more clear. 
  5. There is a lot of missing data. Including information about missing data in the main table makes it difficult to read.
  6. I would suggest providing a more detailed description of the strategy how the problem of missing data was solved in statistical analysis.
  7. I would suggest to provide an extra figure which would present the % of the anwsers somewhat likely, unlikley, very unlikly in the group of vaccine hesistant. 
  8. Why the number of missing data in the case of answer about vaccine hesitancy and continent is the same? Is this only a coincidence? 
  9. In table 1 the Europeans are the most vaccine-hesitant, in the further part of the article you are providing the information that Europeans were more likely to vaccinate comparing with North America (page 7, line 22). This is unclear to me
  10. Instead of "flu vaccine" I suggest the term "influenza vaccination"
  11. In discussion part page 9, line 110 please provide more information what is the influence of influenza vaccination on COVID-19 vaccine hesitancy. There are two, recently published articles that present consistent data to yours. 
    https://www.mdpi.com/2076-393X/9/2/128
    https://www.mdpi.com/2076-393X/9/3/218

    Best Regards
